# Virtual reality as a tool for balance research: Eyes open body sway is reproduced in photo-realistic, but not in abstract virtual scenes

Lorenz Assländer[1]*, Stephan Streuber[1,2]

**1** Universität Konstanz, Konstanz, Germany, **2** Zukunftskolleg der Universität Konstanz, Konstanz, Germany

\* lorenz.asslaender@uni-konstanz.de

**Data Availability Statement:** All relevant data are within the paper and its Supporting Information files.

## Abstract

Virtual reality (VR) technology is commonly used in balance research due to its ability to simulate real world experiences under controlled experimental conditions. However, several studies reported considerable differences in balance behavior in real world environments as compared to virtual environments presented in a head mounted display. Most of these studies were conducted more than a decade ago, at a time when VR was still struggling with major technical limitations (delays, limited field-of-view, etc.). In the meantime, VR technology has progressed considerably, enhancing its capacity to induce the feeling of presence and behavioural realism. In this study, we addressed two questions: Has VR technology now reached a point where balance is similar in real and virtual environments? And does the integration of visual cues for balance depend on the subjective experience of presence? We used a state-of-the-art head mounted VR system and a custom-made balance platform to compare balance when viewing (1) a real-world environment, (2) a photo-realistic virtual copy of the real-world environment, (3) an abstract virtual environment consisting of only spheres and bars ('low presence' VR condition), and, as reference, (4) a condition with eyes closed. Body sway of ten participants was measured in three different support surface conditions: (A) quiet stance, (B) stance on a sway referenced surface, and (C) surface tilting following a pseudo-random sequence. A 2-level repeated measures ANOVA and PostHoc analyses revealed no significant differences in body sway between viewing the real world environment and the photo-realistic virtual copy. In contrast, body sway was increased in the 'low presence' abstract scene and further increased with eyes closed. Results were consistent across platform conditions. Our results support the hypothesis that state of the art VR reached a point of behavioural realism in which balance in photo-realistic VR is similar to balance in a real environment. Presence was lower in the abstract virtual condition as compared to the photo-realistic condition as measured by the IPQ presence questionnaire. Thus, our results indicate that spatial presence may be a moderating factor, but further research is required to confirm this notion. We conceive that virtual reality is a valid tool for balance research, but that the properties of the virtual environment affects results.

**Funding:** LA, SS Zukunftskolleg "Cooperative Initiatives" Universität Konstanz; https://www.uni-konstanz.de/zukunftskolleg/.

**Competing interests:** The authors have declared that no competing interests exist.

## Introduction

Balance control is important for every-day behaviors such as walking, standing, social interactions, dancing, or sports. During these activities, the nervous system controls the body's center of mass counteracting external perturbations, such as gravity [1]. Age and various pathologies such as Parkinson's disease or stroke can deteriorate balance control [2] leading to increased risk of falls and injury [3, 4]. Standing balance is often used as an indicator for these pathologies [5, 6]. Studying balance control is important for understanding its underlying mechanisms and to develop tailored interventions [7, 8]. Conventional experimental paradigms used to study balance control—especially sensory integration mechanisms—often require costly technical equipment such as motion platforms or tilt rooms. These facilities are not accessible to most researchers and clinicians. Here, we test the use of consumer based Virtual Reality (VR) systems as an efficient low-cost alternative for the study of balance control.

Virtual Reality is the presentation of a computer-generated environment that translates the user's behaviors and actions into sensory experiences by replacing sensory feedback using display technologies [9]. In case of head mounted display VR systems (HMDs), stereoscopic images of the virtual scene are rendered from an egocentric viewing perspective obtained from the user's head position and orientation in real time. Due to its ability to simulate rich and vivid real-world experiences, VR presented through HMDs has become a promising methodological tool for studying human perception and behavior [10–12]. Unlike traditional experimental paradigms, VR embeds users into a synthetic computer-generated surrounding. Researchers can use VR to simulate real world experiences and, at the same time, precisely control and manipulate perceptual information. These features, unique to VR, open the possibility to conduct behavioral and psychophysical experiments under more natural and yet controlled experimental conditions, enhancing the ecological validity of the results [13].

A key component of VR is the concept of immersion and presence. Immersion refers to the capacity of a VR system to induce an "inclusive, extensive, surrounding and vivid illusion of reality" [14]. Hence, the degree of immersion mainly depends on technical properties of a VR system such as a large field-of-view, high resolution or multi-sensory stimulation. Presence, on the other hand, is a psychological construct that refers to the subjective experience or illusion of 'being there' in the virtual environment [14]. Presence is (among other factors) affected by visual properties of the environment where photo-realistic virtual environments lead to higher presence scores compared to rather abstract renderings [15]. High degrees of immersion and presence lead to behavioral realism—users respond similarly in virtual and real environments [16]. This makes VR a powerful tool to study human behavior. One specific application is the experimental analysis of the visual contribution to balance control.

Vision is one of three sensory systems used by the central nervous system (CNS) to maintain balance during upright stance. The others being the vestibular system and the proprioceptive reference to the support surface [1]. State of the art HMD VR-technology allows researchers to manipulate the visual scene almost without limitations, providing unique experimental access to the sensory integration of visual cues. However, open questions remain: Is balance viewing a virtual scene comparable to balance viewing a real-world scene? Does the visual contribution to balance depend on subjective experience of the virtual environment (i.e. immersion, presence, etc.)?

Vision research has a long history of using simulated visual inputs and investigating body sway behavior in dependence on different scene presentations. Three different technologies should be distinguished: 2D screen projections [17–20], 3D projections on a 2D screen using stereo-goggles [21–23], and head mounted displays (HMDs). While the difference between 2D and 3D scenes is obvious, 3D visualization on a screen has some fundamental differences to

3D visualization on a HMD [24]. HMDs require online updating of the display when the head is moving, which induces time delays and errors between the head movement and the shift of the scene, however provide a full 3D environment not restricted to a screen. One major problem is the conflict between vergence of the eyes and accommodation of the lense that is present in all 3D environments. This conflict was shown to affect the integration of visual cues for balance control [25].

Very few studies have compared balance behavior in real and virtual environments using HMDs. Two studies found little or no improvement relative to eyes closed spontaneous sway when viewing a stationary virtual visual scene [26], independently from field of view limitations of HMDs [24]. However, VR technology is evolving rapidly, increasing the field of view, optical resolution and accuracy of tracking the HMDs, as well as reducing screen update latencies and device weight. Only one study compared real and virtual environments using a more recent device Robert et al. [27]. These authors used a filmed 3D representation of a laboratory room as the VR environment. Two balance measures were compared: 1) spontaneous sway, i.e. the small body oscillations present during unperturbed upright stance, and 2) more dynamic tasks from the Berg balance scale, such as standing up from a chair or standing one foot, which are subjectively rated from zero (unable) to four (independent) [28]. No differences in performance were found between virtual and real world scenes. The findings of Robert et al. [27] suggest that technology may have evolved to a level where the visual input to balance in virtual reality has similar effects as compared to viewing a real-world visual scene. However, the evidence is still scarce.

In balance research, external perturbations such as support surface tilts are frequently applied in experiments to probe the control mechanism [1]. The relation between perturbation and sway response provides detailed information on the control mechanism maintaining balance, exceeding the information content of spontaneous sway and other purely observational measurements [29]. Therefore, we used support surface tilts to perturb balance in addition to spontaneous sway measurement and compared sway behavior in two stationary virtual scenes to eyes closed and eyes open viewing a real-world scene. The first goal of this study is to test whether balance control is similar in VR and in the real world. If balance control is similar in VR and the real world, we expect similar sway response pattern when participants see a real-world environment compared to when they see a photo-realistic rendering of the same environment in VR, but we expect different sway response patterns when seeing a real-world environment or a photo-realistic rendering in VR as compared to standing with the eyes closed. The second aim of this study was to test the effect of visual richness of the virtual scene on balance control. We hypothesized that different levels of visual richness would modulate the subjective experience of presence, which in turn would have an impact on balance control. Specifically, we hypothesized that a high level of visual richness (e.g. a photo-realistic rendering) would cause an increased feeling of presence leading to similar sway responses in the VR as compared to the real world. In a similar way we expect low feelings of presence for the scene with low visual richness (e.g., impoverished, abstract scenes) leading to deteriorated balance control similar to standing with the eyes closed.

## Methods

### Subjects and ethics statement

Ten young and healthy subjects (25.4 ± 4.7 years; 171 ± 11 cm; 65.5 ± 12.0 kg; 5 female, 5 male) participated in the study after giving written informed consent. Exclusion criteria were orthopedic problems, balance related problems, concussions, and a history of epilepsy. Subject were recruited via an online platform where mostly University students are registered. The

study was conducted in agreement with the declaration of Helsinki in its latest revision and was approved by the University of Konstanz ethics board (IRB20KN07-004).

## Experimental setup

Subjects were wearing a HMD (VIVE PRO EYE, HTC Corporation, Taoyuan City, Taiwan) with a resolution of 1440 x 1600 pixels per eye, a diagonal field of view of 110 degrees. The stereoscopic stimulus was rendered on a Nvidia GeForce RTX 2070 graphics card with 8GB of GDDR with an update frequency of 90 Hz. During the experiment participants stood on a custom-built device with a tiltable support surface, sway-rods to measure body sway, and a force sensor to measure the torque applied at the ankle joints (equivalent to center of pressure measurements). The tilt axis was 8.8 cm above the support surface, thus approximately through the ankle joint axis (Fig 1C). Body sway was measured using two sway rods connected to potentiometers on one end and guided by hooks attached at hip and shoulder level. The torque cues were measured with a force sensor mounted 30 cm below the rotation axis of the tilt platform. With ankle joint and platform axes aligned, ankle torque generated by the subjects resulted in an proportional reaction force at the force sensor. This measurement was only used when the platform was stationary, i.e. when no platform acceleration contributed to the torque measurement. A real-time PC (Education real-time target machine, Speedgoat GmbH, Liebefeld, Switzerland) running Simulink (The Mathworks, Natick, USA) was used to control the device and record the data from the balance device at 1000 Hz sampling rate.

## Visual conditions

Body sway was measured in 4 different visual conditions. Eyes open with a view of the real lab space (EO), a virtual reconstruction of the lab wearing the HMD (LAB, Fig 1A), an abstract

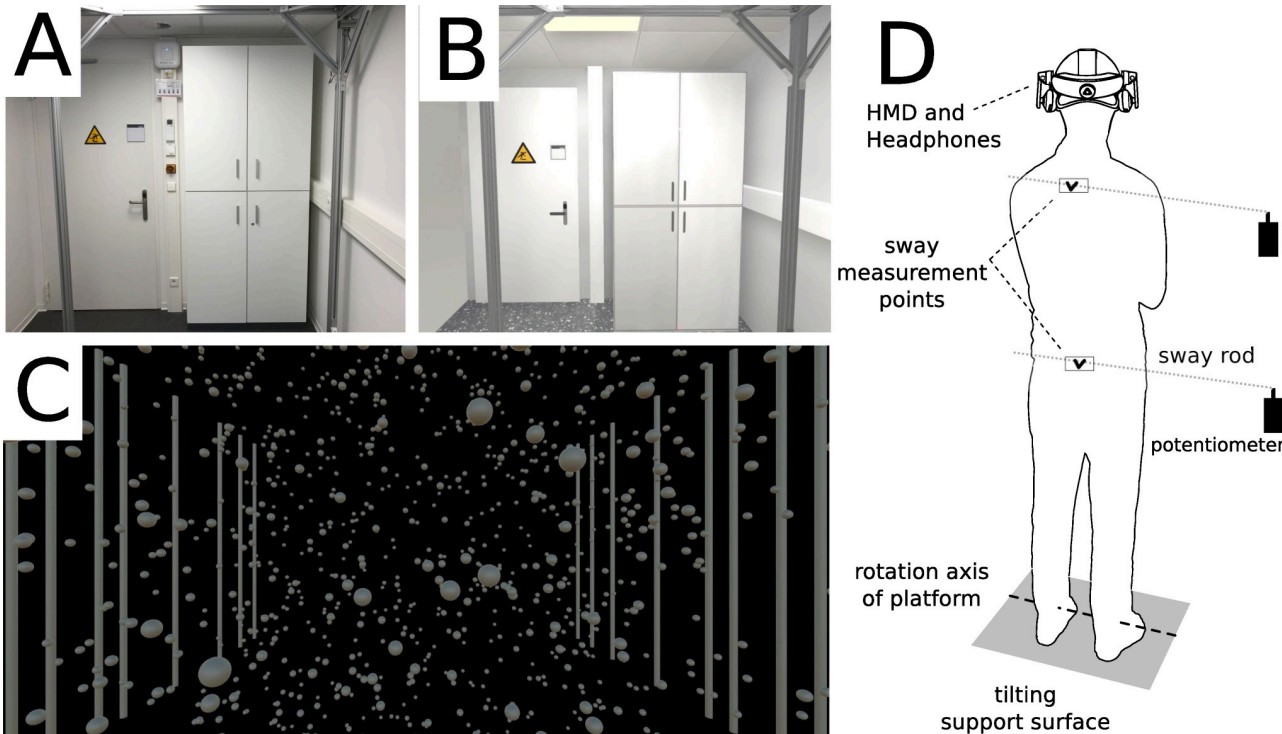

**Fig 1. Experimental setup.** (A) Virtual laboratory scene (LAB). The scene is a detailed reconstruction of the real world view (EO). (B) Virtual abstract scene (ABS). (C) scheme of the experimental setup. Subjects did not wear the HMD in conditions EO and EC.

virtual scene containing a cloud of spheres, as well as vertical bars in the periphery providing rich optic flow information and depth cues wearing the HMD (ABS, Fig 1B), and eyes closed without wearing the HMD (EC). The virtual laboratory scene was 3D modeled in Blender and Unity3D based on measurements and photographs taken from the real laboratory environment. Subjects' presence in and potential nausea from wearing the virtual reality system were assessed following each of the two virtual visual conditions using the igroup Presence Questionnaire (IPQ; [30]) and the Simulator Sickness Questionnaire (SSQ; [31]), respectively. The IPQ was conducted with a scale from 1–5 instead of the recommended scale from 1–6.

## Platform conditions

Three different platform conditions were tested for each visual condition: 1) fixed support surface, 2) sway referenced support surface, and 3) surface tilting in a pseudo-random sequence.

For the *fixed surface*, sway was measured for 45 seconds. *Sway referencing* is a condition, where the platform is tilting with the subject [32]. Sway referencing is achieved by measuring the tilt of a subjects' leg segment using the hip sway rod and online calculation and control of the platform tilt at each sample time using the real-time PC. The calculation is tuned such that swaying forward and backward does not change the ankle angle as it would on a fixed surface. Therefore, the proprioceptive reference to the support surface contains virtually no information about body orientation in space and is thus not contributing to balance control, increasing the contribution of visual and vestibular inputs [1]. Body sway during sway referencing was measured for 45 seconds. The sway referencing condition immediately followed the quiet stance condition (i.e. both were measured in the same trial). For the *surface tilt* condition, the tilt stimulus was based on a 20-s long pseudo-random-ternary sequence [1]. The sequence consists of eighty 0.25-s long states with positive (+1.78°/s), negative (-1.78°/s) or zero velocity, constructed in pseudo-random order using shift-registers [33]. The platform tilt angle followed the integrated signal, which had a peak-to-peak tilt amplitude of 4°. The subjective experience during these pseudo-random stimuli are small toes-up and toes-down movements with unpredictable changes in direction. A total of 13 sequences were concatenated, resulting in a trial length of 260 s.

## Center of mass calibration routine

A 120 second long calibration routine was used to obtain anterior-posterior whole body center of mass (com) position without assumptions on segment mass distributions [1]. Subjects were instructed to make very slow movements in the ankle and hip joints within the movement range typically seen during the experiment. In these quasi-static conditions, the center of pressure position ($x_{cop}$), obtained from the torque measurement, can be used as a projection of the com position ($x_{com}$; [34]:

$$x_{cop}(t) \approx x_{com}(t) \qquad (1)$$

A least-squares regression between hip and shoulder translation, obtained from the sway rods and trigonometric calculations, and the center of pressure were used to calculate calibration values. In conditions with larger and more dynamic body sway, where Eq 1 does not hold anymore, these calibration values can be used to calculate the com position based on the hip and shoulder movements:

$$x_{com}(t) = a + b*x_{hip}(t) + c*x_{sho}(t) \qquad (2)$$

Finally, using an estimate of the body com height ($h_{com}$; [35], angular body com sway ($\theta_{com}$) was calculated and used for all further analysis.

$$\theta_{com}(t) = \arcsin \left( x_{com}(t)/h_{com} \right) \tag{3}$$

## Procedures

After obtaining written informed consent, anthropometric measures were taken from subjects. Subjects briefly familiarized with the HMD and both virtual scenes (LAB, ABS) in a seated position prior to the experiments. Subjects were then positioned on the platform and sway rods were attached. After the calibration routine, subjects familiarized with the different tilt stimulus conditions and the virtual lab scene (LAB) during a 150-s long warm-up session. We confirmed that subjects were comfortable with the setup, before running visual conditions EO, LAB, ABS, and EC in randomized order. Within each visual condition, the two stimulus sequences (pseudo-random tilt and quiet stance with fixed surface followed by sway referenced support surface) were run in randomized order. During all experimental conditions including the warmup, subjects wore noise-canceling head-phones listening to audio books or podcasts to avoid auditory orientation cues and distract from the balance task. Following either of the two virtual reality conditions, subjects were asked to verbally answer the questionnaires (SSQ and IPQ).

## Data analysis and statistics

Recorded data was analyzed using Matlab (The Mathworks, Natick, USA). For quiet stance and sway referencing trials, com trajectories were low-pass filtered to remove measurement noise ($2^{nd}$ order butterworth filter; cutoff 5 Hz) and the first 15 seconds were discarded. For the remaining 30 seconds, anterior-posterior sway path was calculated by summing the absolute sample-to-sample difference. This measure of the sway path was then divided by trial time to obtain a measure of average sway velocity (s) for each subject, which was used for statistical analysis.

For pseudo-random surface tilt trials, the first of the thirteen sequences was discarded to avoid transients at the beginning of the tilt perturbations and each cycle was centered around zero (subtracting the mean). The arithmetic mean and the standard deviation across the remaining 12 sequences were calculated for each subject to obtain a periodic sway component as a measure of the response to the tilt stimulus and the random sway component not evoked by the stimulus, respectively. Sway power of the periodic and the random component was calculated as the sum of the squared trace.

JASP [36] was used for all statistical analyses. Parameters were statistically compared using a two level repeated measures ANOVA with the levels 'visual condition' and 'platform condition'. Further, differences of visual conditions with respect to eyes open were tested using simple contrasts. PostHoc comparisons of visual conditions within each platform condition were calculated using single level repeated measures ANOVAs and simple contrasts with respect to EO.

## Results

One subject showed extraordinarily large sway in all platform conditions. The data of this subject is marked red in the plots and is not included in mean values and statistical analyses. Table 1 shows the results of the repeated measures ANOVA. Visual condition, platform condition and their interaction showed significant differences ($p < .001$). Table 2 shows the contrasts comparing LAB, ABS, and EC to EO. There was no significant difference between EO and the virtual reality condition LAB, however there was a difference for ABS and EC ($p <$

**Table 1. Within subjects differences between platform and visual conditions.**

| | Sum of Squares | df | Mean Square | F | p | $\eta^2$ |
|---|---|---|---|---|---|---|
| platform condition | 3.136 | 3 | 1.045 | 53.775 | < .001 | 0.526 |
| Residual | 0.467 | 24 | 0.019 | | | |
| visual condition | 0.914 | 3 | 0.305 | 35.317 | < .001 | 0.153 |
| Residual | 0.207 | 24 | 0.009 | | | |
| visual condition * platform condition | 0.518 | 9 | 0.058 | 11.929 | < .001 | 0.087 |
| Residual | 0.348 | 72 | 0.005 | | | |

*Note*. repeated measures ANOVA; Type III Sum of Squares.

*.001*). In the following, we will describe the changes for individual platform conditions and the questionnaire results.

## Surface tilt conditions

Fig 2 shows the results of the support surface tilt conditions with the actual platform movement for one 20 second pseudo-random sequence (Fig 2A) and periodic body sway (averaged across 108 sequence repetitions; 9 subjects à 12 sequences) in the time domain (Fig 2B). Periodic body sway reflects the general shape of the stimulus sequence and shows small differences between EO, LAB and ABS, and a big difference between EO and EC. The power of sway responses to the tilt stimulus and sway variability during the tilt conditions are shown in Fig 3. Sway power of the periodic component was 2–4 times larger as compared to that of the random sway component. The ANOVA showed significant differences in visual conditions ($p <$ *.001*). Contrasts showed no significant difference between EO and LAB for periodic sway ($p =$ *.450*) or variability ($p =$ *.258*). Both parameter differed from EO for ABS and EC (p < .001).

## Spontaneous sway conditions

Fig 4 shows the results of the two conditions in which spontaneous sway was measured: subjects standing on a fixed surface (left) and on a sway referenced surface (right). Sway power during sway referenced conditions was about 3 times larger as compared to the fixed surface conditions. The ANOVA again showed an effect of visual conditions. Contrasts showed no difference between EO and LAB for fixed surface ($p =$ *.506*) and sway referenced surface ($p =$ *.498*). There was no significant difference between EO and ABS on a fixed surface ($p =$ *.083*), however there was a difference on the sway referenced surface ($p <$ *.01*). EC differed significantly from EO ($p <$ *.001*).

## Questionnaire results

Table 3 shows the average results of the SSQ and the IPQ across all subjects. The SSQ showed extremely low values, indicating that subjects had no nausea problems in the virtual

**Table 2. Post-hoc simple contrasts comparing visual conditions.**

| Comparison | Estimate | SE | t | p |
|---|---|---|---|---|
| LAB—EO | 0.023 | 0.022 | 1.043 | 0.307 |
| ABS—EO | 0.105 | 0.022 | 4.779 | < .001 |
| EC—EO | 0.203 | 0.022 | 9.275 | < .001 |

*Note*. EO real world scene; EC eyes closed; LAB virtual laboratory scene; ABS virtual abstract scene.

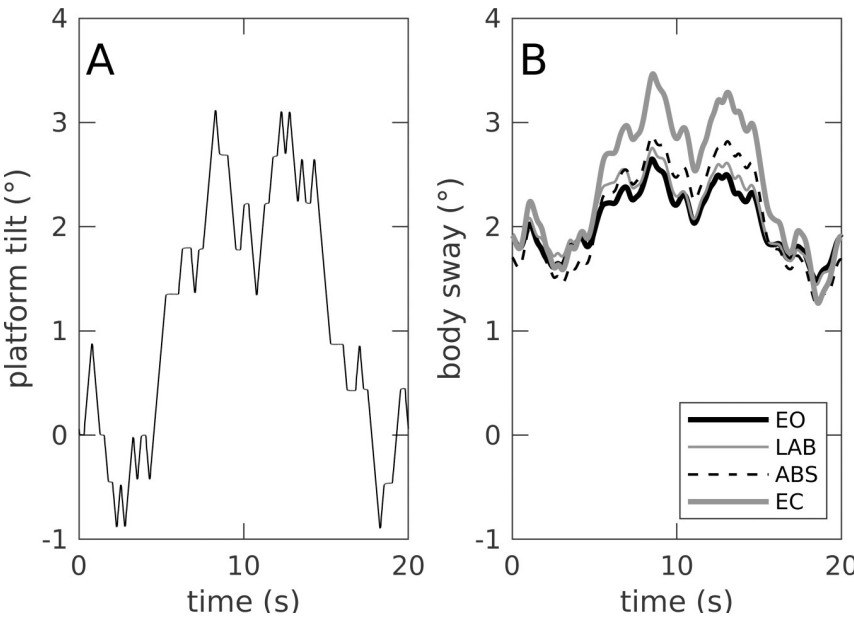

**Fig 2. Example sequences support surface tilt and periodic body sway.** (A) Support surface tilt sequence (was repeated 13 times in each trial) and (B) periodic body com sway in response to the stimulus (= average across sequence repetitions and subjects) for all visual conditions. Visual conditions are real scene eyes open (EO), virtual laboratory room (LAB), virtual abstract scene (ABS), and eyes closed (EC).

environment. The IPQ showed a larger presence (P) in the LAB scene, which was reflected by slightly larger values for the three sub categories spatial presence (SP), involvement (INV) and experienced realism (REAL).

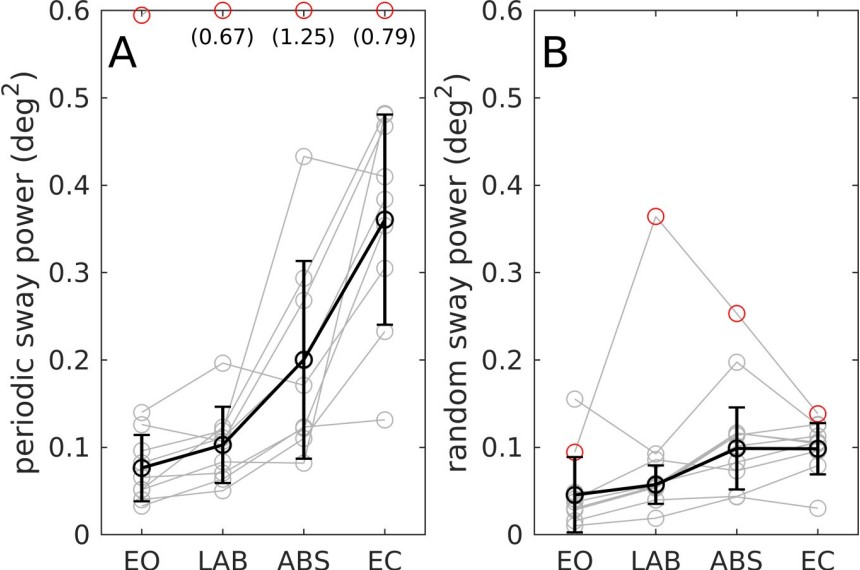

**Fig 3. Body sway power during pseudo-random platform tilts.** (A) Periodic sway component. (B) Random sway component. Visual conditions are real scene eyes open (EO), virtual laboratory room (LAB), virtual abstract scene (ABS), and eyes closed (EC). Parameters are shown for individual subjects (grey) and as mean and standard deviation across subjects (black). Red circles indicate outliers (one subject) not included in the average. Values outside the visible range are given in brackets.

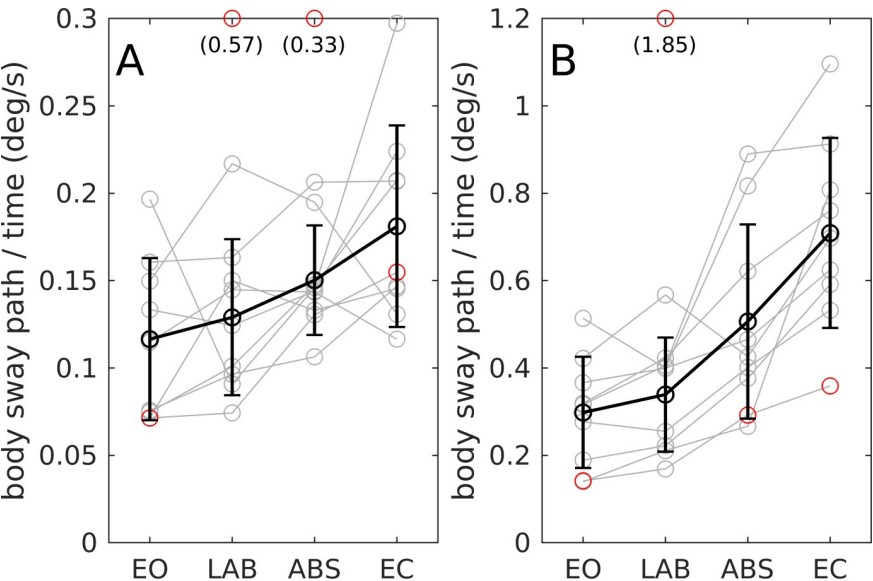

**Fig 4. Mean body sway velocity.** (A) Fixed support surface. (B) Sway referenced support surface. Visual conditions are real scene eyes open (EO), virtual laboratory room (LAB), virtual abstract scene (ABS), and eyes closed (EC). Sway velocity is shown for individual subjects (grey circles) and as mean and standard deviation across subjects (black). Red circles indicate outliers (one subject) not included in the average. Values outside the visible range are given in brackets.

## Discussion

The results showed a very consistent behavior across all support surface conditions. Subjects showed very similar sway viewing the virtual laboratory scene (LAB) as compared to the real laboratory scene (EO). However, sway was larger when viewing the abstract virtual scene (ABS). This increase was about 50% of the increase found for eyes closed (EC) as compared to eyes open conditions. The similarity of EO and LAB conditions shows that human subjects use visual inputs also in a virtual environment as a sensory input to maintain balance. However, the difference in sway when viewing the abstract scene indicates that humans do not use every virtual scene alike. Below, we discuss two possible factors that may modulate the visual contribution to standing balance.

### Presence

Improved balance in the photo-realistic laboratory scene was associated by higher presence scores in the igroup presence questionnaire as compared to the abstract scene. This finding is in line with research showing 1) that photo-realistic virtual environments lead to higher presence scores compared to rather abstract renderings [15] and 2) that a high presence score is a predictor of behavioral realism. Thus, subjects might have used the visual reference in the abstract scene less as compared to the photo-realistic laboratory scene due to different levels of

**Table 3. Questionnaire results.**

| VR scene | SSQ | IPQ G | IPQ SP | IPQ INV | IPQ REAL |
|---|---|---|---|---|---|
| LAB | 7.5 ± 8.5 | 3.7 ± 0.9 | 3.5 ± 0.6 | 3.2 ± 0.5 | 2.7 ± 0.2 |
| ABS | 15.3 ± 14.8 | 2.6 ± 1.4 | 3.3 ± 0.8 | 3.1 ± 0.8 | 2.1 ± 0.3 |

*Note*. SSQ max Score: 235.62; IPQ Score: 1–5.

presence. Such a possible relationship between presence and postural stability has been suggested by Menzies et al. [37]. The authors observed small differences when comparing spontaneous sway viewing the same virtual scene with different technical devices. Menzies et al. [37] found reduced sway using a better technical device and dedicated their findings to higher fidelity. While their findings are in line with our results and interpretation, Menzies et al. [37] used a visual scene similar to our abstract scene. Balance measures were overall close to eyes closed behavior and very different from eyes open behavior, which was nicely reproduced in our abstract scene results. Thus, the findings of Menzies et al. [37] may have been much more clear when using a more realistic virtual environment. Nonetheless, the findings of Menzies et al. [37] and our findings support the idea that postural stability might be a predictive and unbiased behavioral marker for presence. However, further research needs to be conducted to better quantify this relationship.

## Cognitive suppression of the visual contribution

One striking aspect of our results is that balance in the abstract virtual scene was only slightly improved as compared to eyes closed, despite a supposedly rich visual input (large field optic flow, depth and parallax information, etc.) and using the same technical device. Above we argue that the difference to the virtual lab scene might be related to the subjective feeling of presence in the scene. However, it is unclear how a feeling of presence could affect the balance control mechanism. One mechanism could be a cognitive suppression of the visual input. It is well known that humans can suppress the visual input to standing balance. For example, Bronstein [38] showed that subjects show a large sway response when exposed to a sudden movement of a (real world) visual surround for the first time. However, the sway response is largely suppressed during a second and succeeding exposures. The relevance of this suppression mechanism is very straight forward: subjects would fall when looking at a driving bus, if the input could not be suppressed. A similar suppression could also occur when subjects do not feel present in the scene. In other words, the feeling of presence in the virtual environment could modulate the extend to which a subject 'trusts' and therefore uses the visual scene as a reliable visual space reference for standing balance. Such a causal relation of presence and a cognitive suppression of the visual contribution to balance is plausible, but remains speculative at this stage.

## Information content of the visual input

The central nervous system extracts a variety of different visual cues from the retinal input, such as optic flow, visual vertical, various depth cues, or landmark information. The abstract scene was constructed to contain a rich visual input. However, the visual input may have not provided the same amount or quality of information in the abstract environment, as compared to the virtual laboratory room. For example, the abstract scene did not contain a floor or walls, which may provide a much stronger self-orientation information as compared to the vertical bars in the periphery.

## Limitations

In VR balance control, mechanisms might be challenged by technical artifacts such as a limited field of view or delays. A lack of balance control in VR due to these factors might also lead to discomfort, cybersickness, falls or injury. The restricted field of view using the HMD may have led to an increase in sway responses [39]. Close inspection showed that sway in the laboratory scene (LAB) was slightly larger as compared to sway with eyes open in the real-world (EO). Thus, the limited field of view of the HMD may have reduced the visual information content

to some extent. However, overall the effect was small and, as discussed above, could also have been caused by other reasons.

Overall, the study showed very clear results, despite the small number of subjects. Nonetheless, investigating the above discussed relation of presence and the visual contribution to standing balance requires a much larger power and needs to resolve the problem of providing the same visual information content in low and high presence conditions.

In conclusion, we found that human subjects use visual self-motion cues for standing balance in a virtual environment. However, the visual input is reduced in an abstract scene, where we identified a reduced presence in the virtual environment or a poorer quality of the visual self-motion cues as potential reasons. Our findings indicate that virtual reality is a useful tool for balance research. In addition, balance behavior may also be a good tool to quantify presence of a subject in the virtual environment.

## Supporting information

**S1 Data.**
(ZIP)

## Acknowledgments

We would like to thank Amine El-Kaissi, Sandra Wacker, Sharan Gopalan, and Nikolai Killer for their help creating the scenes and during data collection.

## Author Contributions

**Conceptualization:** Lorenz Assländer, Stephan Streuber.

**Data curation:** Lorenz Assländer, Stephan Streuber.

**Formal analysis:** Lorenz Assländer.

**Funding acquisition:** Lorenz Assländer, Stephan Streuber.

**Investigation:** Lorenz Assländer, Stephan Streuber.

**Methodology:** Lorenz Assländer, Stephan Streuber.

**Project administration:** Lorenz Assländer, Stephan Streuber.

**Resources:** Lorenz Assländer, Stephan Streuber.

**Software:** Lorenz Assländer, Stephan Streuber.

**Supervision:** Lorenz Assländer, Stephan Streuber.

**Validation:** Lorenz Assländer, Stephan Streuber.

**Visualization:** Lorenz Assländer, Stephan Streuber.

**Writing – original draft:** Lorenz Assländer.

**Writing – review & editing:** Lorenz Assländer, Stephan Streuber.

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
