## [Decision Letter · Decision Letter 0]

11 Sep 2020

PONE-D-20-23613

Virtual reality as a tool for balance research: eyes open body sway is reproduced in photo-realistic, but not in abstract virtual scenes

PLOS ONE

Dear Dr. Assländer,

Thank you for submitting your manuscript to PLOS ONE. After careful consideration, we feel that it has merit but does not fully meet PLOS ONE’s publication criteria as it currently stands. Therefore, we invite you to submit a revised version of the manuscript that addresses the points raised during the review process.

We look forward to receiving your revised manuscript.

Kind regards,

Eric R. Anson

Academic Editor

PLOS ONE

Journal Requirements:

2. Please ensure you have thoroughly discussed all potential limitations of this study within the Discussion section, including the small sample size.

3.Thank you for including your ethics statement:  "The  study  was  conducted  in  agreement with  the  declaration  of  Helsinki  in  its  latest  revision  and  in  agreement  with  the  University  of Konstanz ethics regulations".   

Please amend your current ethics statement to confirm that your named institutional review board or ethics committee specifically approved this study.

Reviewers' comments:

Reviewer's Responses to Questions

**Comments to the Author**

1. Is the manuscript technically sound, and do the data support the conclusions?

Reviewer #1: Yes

Reviewer #2: Yes

2. Has the statistical analysis been performed appropriately and rigorously? 

Reviewer #1: Yes

Reviewer #2: Yes

3. Have the authors made all data underlying the findings in their manuscript fully available?

Reviewer #1: Yes

Reviewer #2: Yes

4. Is the manuscript presented in an intelligible fashion and written in standard English?

Reviewer #1: Yes

Reviewer #2: Yes

5. Review Comments to the Author

Reviewer #1: General comments

This manuscript is well written and concerns a highly interesting topic, since VR is use increasingly in rehabilitation. The manuscript is well written. Please find my specific comments below.

Title

Explicit

Abstract

Clear

Introduction

Appropriate.

Aim

Long, but clear

Method

How were the subjects recruited? Were they healthy subjects, students, staff? How many women and men? Differences in response to immersive virtual reality environment has been found between men and women in other studies.

Is there any approval from an ethical review board?

Results

Appropriate

Discussion

Line 341 – 348, “Cognitive suppression of the visual contribution”. The paragraph is interesting but needs to be in relation to the findings in this study.

Only limitations of balance in VR are discussed. Please add a paragraph with strengths and limitations of the study.

Conclusion

Well written

Tables and figures

Table 1 shows within subjects differences in the different conditions, please clarify this in the heading. Also include the statistical analysis used when calculating the differences, preferably in a footnote.

Table 2 also needs a clarifying heading. Abbreviations should be explained in a footnote.

References

Appropriate

Reviewer #2: Virtual reality as a tool for balance research: eyes open body sway is reproduced in photo-realistic, but not in abstract virtual scenes

The study compared the amount of postural control (body sway) between reconstructed VR environment (LAB scene) and the real world, together with comparison against other visual conditions. Authors found that postural sway was in the real world was synonymous to the LAB scene but not with eyes closed or abstract virtual scenes. They concluded that VR is a valid tool for conducting balance research. The study was well performed, well written to clearly communicate the results and has added additional evidence for the usage of VR in balance research. There are, however, a few grammatical considerations and comments needed.

Introduction

Line 39: The “Who” reference should read “WHO”

Line 146: Change “was” to “of”.

Line 146: Rephrase to clarify meaning. Check word(s) omitted in sentence beginning with “During the experiment…”, specifically, “…were while…” and “…instrumentation to measure…”

Line 172: Check word(s) omitted in “from were”.

Discussion

Lines 330 – 339: Conducting further research to unravel why inconsistent results were found in the Menzies’ study is in order. However, the present authors could compare the nature of the visual scenes in the both studies to provide some possible reasons. For instance, the LAB scene may have resulted to a lesser postural instability compare to the ABS scene because it was a reconstruction of the real world. It is clear that this LAB scene is different from the visual disturbances in the Menzies’ study.

Line 344: “(Bronstein, 1986)” should be written as “Bronstein (1986)”.

6. PLOS authors have the option to publish the peer review history of their article (what does this mean?). If published, this will include your full peer review and any attached files.

Reviewer #1: No

Reviewer #2: **Yes: **Kwadwo Osei Appiah-Kubi

---

## [Author Response · Author response to Decision Letter 0]

14 Sep 2020

We would like to thank both reviewers for the positive feedback and their helpful remarks. Below is a point-by-point response to the comments.

Reviewer #1: General comments

This manuscript is well written and concerns a highly interesting topic, since VR is use increasingly in rehabilitation. The manuscript is well written. Please find my specific comments below.

Title

Explicit

Abstract

Clear

Introduction

Appropriate.

Aim

Long, but clear

Method

How were the subjects recruited? Were they healthy subjects, students, staff? How many women and men? Differences in response to immersive virtual reality environment has been found between men and women in other studies.

Is there any approval from an ethical review board?

 We specified the open questions in the manuscript.

 We rephrased the ethics statement, now explicitly refering to the approval.

New paragraph: „Ten young and healthy subjects (age 25.4 ± 4.7 years; height 171 ± 11 cm; weight 65.5 ± 12.0 kg; 5 female, 5 male) participated in the study after giving written informed consent. Exclusion criteria were orthopedic problems, balance related problems, concussions, and a history of epilepsy. Subject were recruited via an online platform where mostly University students are registered. The study was conducted in agreement with the declaration of Helsinki in its latest revision and in agreement with was approved by the University of Konstanz ethics board (IRB20KN07-004).“

Results

Appropriate

Discussion

Line 341 – 348, “Cognitive suppression of the visual contribution”. The paragraph is interesting but needs to be in relation to the findings in this study.

 Thank you for pointing out that we did not describe our point properly. We clarified our statement and put it into context.

Only limitations of balance in VR are discussed. Please add a paragraph with strengths and limitations of the study.

 We generalized the paragraph and added several aspects, such as the small number of subjects (limitation) and the strong effects we found in our results (strength).

Conclusion

Well written

Tables and figures

Table 1 shows within subjects differences in the different conditions, please clarify this in the heading. Also include the statistical analysis used when calculating the differences, preferably in a footnote.

Table 2 also needs a clarifying heading. Abbreviations should be explained in a footnote.

 Amended as suggested.

References

Appropriate

Reviewer #2: Virtual reality as a tool for balance research: eyes open body sway is reproduced in photo-realistic, but not in abstract virtual scenes

The study compared the amount of postural control (body sway) between reconstructed VR environment (LAB scene) and the real world, together with comparison against other visual conditions. Authors found that postural sway was in the real world was synonymous to the LAB scene but not with eyes closed or abstract virtual scenes. They concluded that VR is a valid tool for conducting balance research. The study was well performed, well written to clearly communicate the results and has added additional evidence for the usage of VR in balance research. There are, however, a few grammatical considerations and comments needed.

Introduction

Line 39: The “Who” reference should read “WHO”

 Corrected. Thank you!

Line 146: Change “was” to “of”.

 Corrected. Thank you!

Line 146: Rephrase to clarify meaning. Check word(s) omitted in sentence beginning with “During the experiment…”, specifically, “…were while…” and “…instrumentation to measure…”

 We cleaned up the sentence.

Line 172: Check word(s) omitted in “from were”.

 Added. Thank you!

Discussion

Lines 330 – 339: Conducting further research to unravel why inconsistent results were found in the Menzies’ study is in order. However, the present authors could compare the nature of the visual scenes in the both studies to provide some possible reasons. For instance, the LAB scene may have resulted to a lesser postural instability compare to the ABS scene because it was a reconstruction of the real world. It is clear that this LAB scene is different from the visual disturbances in the Menzies’ study.

 We fully agree with this comment and rewrote the paragraph to elaborate on the similarity of our abstract scene and the abstract scene used by Menzies et al. and the difference to our LAB scene. We now also point out that the Menzies findings may have been much more clear when using a more realistic scene.

Line 344: “(Bronstein, 1986)” should be written as “Bronstein (1986)”.

 Corrected. Thank you!

---

## [Decision Letter · Decision Letter 1]

21 Sep 2020

PONE-D-20-23613R1

Virtual reality as a tool for balance research: eyes open body sway is reproduced in photo-realistic, but not in abstract virtual scenes

PLOS ONE

Dear Dr. Assländer,

Thank you for submitting your manuscript to PLOS ONE. After careful consideration, we feel that it has merit but does not fully meet PLOS ONE’s publication criteria as it currently stands. Therefore, we invite you to submit a revised version of the manuscript that addresses the points raised during the review process.

Overall the reviewer responses to the changes made were very positive.  Although much improved, the paragraph "Cognitive suppression of the visual contribution" in the discussion which highlights important elements comes across as too general and disconnected from the results of the study.  In your revision, please more explicitly connect this paragraph to the study results while also maintaining the broader context that is currently presented.

We look forward to receiving your revised manuscript.

Kind regards,

Eric R. Anson

Academic Editor

PLOS ONE

Journal Requirements:

Additional Editor Comments (if provided):

Reviewers' comments:

Reviewer's Responses to Questions

**Comments to the Author**

1. If the authors have adequately addressed your comments raised in a previous round of review and you feel that this manuscript is now acceptable for publication, you may indicate that here to bypass the “Comments to the Author” section, enter your conflict of interest statement in the “Confidential to Editor” section, and submit your "Accept" recommendation.

Reviewer #1: (No Response)

Reviewer #2: All comments have been addressed

2. Is the manuscript technically sound, and do the data support the conclusions?

Reviewer #1: Yes

Reviewer #2: Yes

3. Has the statistical analysis been performed appropriately and rigorously? 

Reviewer #1: Yes

Reviewer #2: Yes

4. Have the authors made all data underlying the findings in their manuscript fully available?

Reviewer #1: Yes

Reviewer #2: Yes

5. Is the manuscript presented in an intelligible fashion and written in standard English?

Reviewer #1: Yes

Reviewer #2: Yes

6. Review Comments to the Author

Reviewer #1: Thank you for your revised manuscript, it is much improved. The paragraph in the discussion called “Cognitive suppression of the visual contribution” is, as said before, very interesting but needs to be in relation to the findings of your study. As it is written now, it could preferably be in the introduction. Please revise the paragraph once again to better fit in the discussion.

Reviewer #2: (No Response)

7. PLOS authors have the option to publish the peer review history of their article (what does this mean?). If published, this will include your full peer review and any attached files.

Reviewer #1: No

Reviewer #2: No

---

## [Author Response · Author response to Decision Letter 1]

13 Oct 2020

Reviewer #1: Thank you for your revised manuscript, it is much improved. The paragraph in the discussion called “Cognitive suppression of the visual contribution” is, as said before, very interesting but needs to be in relation to the findings of your study. As it is written now, it could preferably be in the introduction. Please revise the paragraph once again to better fit in the discussion.

→ We are sorry for this unnecessary iteration. We were probably mentally stuck in follow-up experiments and hope we now addressed your justified criticism better. For your convenience we copied the revised paragraph below, which is the only changed section in the manuscript.

Cognitive suppression of the visual contribution

One striking aspect of our results is that balance in the abstract virtual scene was only slightly improved as compared to eyes closed, despite a supposedly rich visual input (large field optic flow, depth and parallax information, etc.) and using the same technical device. Above we argue that the difference to the virtual lab scene might be related to the subjective feeling of presence in the scene. However, it is unclear how a feeling of presence could affect the balance control mechanism. One mechanism could be a cognitive suppression of the visual input. It is well known that humans can suppress the visual input to standing balance. For example, Bronstein (1986) showed that subjects show a large sway response when exposed to a sudden movement of a (real world) visual surround for the first time. However, the sway response is largely suppressed during a second and succeeding exposures. The relevance of this suppression mechanism is very straight forward: subjects would fall when looking at a driving bus, if the input could not be suppressed. A similar suppression could also occur when subjects do not feel present in the scene. In other words, the feeling of presence in the virtual environment could modulate the extend to which a subject ‘trusts’ and therefore uses the visual scene as a reliable visual space reference for standing balance. Such a causal relation of presence and a cognitive suppression of the visual contribution to balance is plausible, but remains speculative at this stage.

---

## [Decision Letter · Decision Letter 2]

16 Oct 2020

Virtual reality as a tool for balance research: eyes open body sway is reproduced in photo-realistic, but not in abstract virtual scenes

PONE-D-20-23613R2

Dear Dr. Assländer,

We’re pleased to inform you that your manuscript has been judged scientifically suitable for publication and will be formally accepted for publication once it meets all outstanding technical requirements.

Kind regards,

Eric R. Anson

Academic Editor

PLOS ONE

Additional Editor Comments (optional):

Reviewers' comments:

Reviewer's Responses to Questions

**Comments to the Author**

1. If the authors have adequately addressed your comments raised in a previous round of review and you feel that this manuscript is now acceptable for publication, you may indicate that here to bypass the “Comments to the Author” section, enter your conflict of interest statement in the “Confidential to Editor” section, and submit your "Accept" recommendation.

Reviewer #1: All comments have been addressed

2. Is the manuscript technically sound, and do the data support the conclusions?

Reviewer #1: Yes

3. Has the statistical analysis been performed appropriately and rigorously? 

Reviewer #1: Yes

4. Have the authors made all data underlying the findings in their manuscript fully available?

Reviewer #1: Yes

5. Is the manuscript presented in an intelligible fashion and written in standard English?

Reviewer #1: Yes

6. Review Comments to the Author

Reviewer #1: Thank you for your revision, all my concerns are met, and the paragraph in the discussion is perfect!

7. PLOS authors have the option to publish the peer review history of their article (what does this mean?). If published, this will include your full peer review and any attached files.

Reviewer #1: **Yes: **Eva Ekvall Hansson

---

## [Editor Report · Acceptance letter]

20 Oct 2020

PONE-D-20-23613R2 

Virtual reality as a tool for balance research: eyes open body sway is reproduced in photo-realistic, but not in abstract virtual scenes 

Dear Dr. Assländer:

I'm pleased to inform you that your manuscript has been deemed suitable for publication in PLOS ONE. Congratulations! Your manuscript is now with our production department. 

Kind regards, 

on behalf of

Dr. Eric R. Anson 

Academic Editor

PLOS ONE